# How to Adapt Your Large-Scale Pre-trained Model for Downstream Image Classification

## Abstract

Pre-training large-scale vision and language models (e.g. CLIP) has shown promising results in representation and transfer learning. We investigate the question of how to efficiently adapt these models to downstream tasks. For image classification, linear probes have been the standard for ease of use and efficiency, while for language, other approaches like prompt tuning have emerged. We analyze several fine-tuning methods across a diverse set of image classification tasks across two spectra investigating the amount and similarity of downstream data to that of pretraining one. We find that just tuning LayerNorm parameters is a surprisingly effective baseline across the board. We further demonstrate a simple yet effective strategy that combines LayerNorm-tuning with general fine-tuning methods to improve their performance and benchmark them on few-shot adaption and distribution shift tasks. Finally, we provide an empirical analysis and recommend general recipes for efficient transfer learning of vision and language models [1].

## 1 Introduction

Large-scale deep network models pretrained on ultra large-scale data on the internet, whether text or images, have shown impressive performance recently (Radford et al., 2019; 2021; Brown et al., 2020; Devlin et al., 2018; Jia et al., 2021). Training such models with billions of parameters on a large internet scale data is an expensive and time consuming process often costing over millions of dollars. Hence, replicating such models is not only difficult but also undesirable for every downstream task. Fortunately, the information gathered by these large-scale models using raw internet data seems to transfer well to several downstream tasks with little to no finetuning at all using natural language as a way for zero-shot evaluation (Brown et al., 2020; Radford et al., 2021).

While zero-shot transfer performs well, it is generally better to adapt the model itself if there are any labeled examples available for the downstream task. Traditionally, the go-to strategy in the computer vision community has either been to finetune the whole network or an additional MLP layer at the end. With the use of raw language, adaptation techniques such as prompt tuning have surfaced (Li & Liang, 2021; Lester et al., 2021). Alternative methods include new parameters in between the network instead of adding a layer at the end (Houlsby et al., 2019; Mahabadi et al., 2021). However, it remains unclear as to which approach is preferred under which scenarios.

We ask what are the general guidelines one should adopt while finetuning a large-scale pretrained model on downstream datasets. To scope this question, we choose CLIP (Radford et al., 2021) as the base pretrained model and adapt it to several downstream problems. CLIP is a vision-and-language model trained on over 400M pairs of image and text descriptions collected off the internet. There are several reasons to choose CLIP for this study. First, CLIP is one of the few vision models trained on ultra-large scale, unfiltered and varied raw visual data on the internet. Second, the multi-modal nature of CLIP enables use of more general ways of adaptation like using natural language prompts for "zero-shot" transfer to new categories – techniques previously popular mostly in NLP.

We find that merely tuning the parameters of LayerNorm (Ba et al., 2016) turns out to be a *surprisingly effective approach* that is competitive or better than all other adaptation methods across the board. The effectiveness of normalization techniques has been observed by prior work for generalization (Perez et al., 2018; Lu et al., 2021) as well as training from scratch (Frankle et al., 2020). Inspired by this,

---

[1]Website at https://sites.google.com/view/adapt-large-scale-models

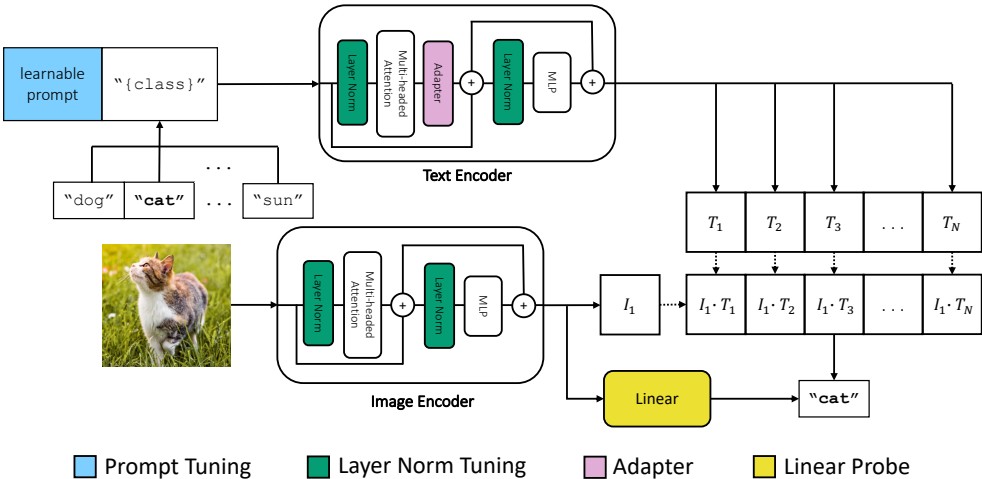

Figure 1: Illustration of multiple methods for adapting CLIP to downstream image classification tasks. Each labeled approach can be used separately for fine-tuning the CLIP model. We analyze a variety of fine-tuning methods such as prompt tuning by prepending a learnable prompt, tuning Layer Normalization parameters, inserting adapter and compacter modules in-between the Transformer layers, and using a linear probe on top of visual features. The CLIP model can be also used for inference on a downstream task in a zero-shot manner.

we further look into different ways of combining LayerNorm-tuning with other adaptation methods that finetune new parameters. We devise an effective scheme that first finetunes the CLIP model using only LayerNorm tuning and uses it as initialization for adapting new parameters. We evaluate our adaptation techniques across 12 downstream tasks spread along two spectra: size of downstream task dataset as well as the similarity of downstream data to the pretraining data. Across both spectra, we find that our two-stage LayerNorm tuning approach is most competitive and show its effectiveness for general-purpose adaptation of CLIP to downstream image-classification tasks.

To summarize, our paper's contributions are as follows:

- We show the effectiveness of LayerNorm-tuning for adaptation to downstream tasks.
- We devise a simple yet effective scheme to combine LayerNorm-tuning with other methods of finetuning to obtain competitive performance across the board.
- We show a thorough comparison of different adaptation methods in four scenarios across two spectra (amount of downstream data and its similarity to pretraining data) studied on numerous downstream classification tasks.

We believe our findings will encourage more research and put existing research in perspective of what works best while finetuning large-scale vision-language models to downstream tasks.

## 2 BACKGROUND: VISION-AND-LANGUAGE PRETRAINED MODELS

Vision-and-language pre-training methods have recently shown promise on diverse tasks across images and text (Radford et al., 2021; Jia et al., 2021). While many such approaches have emerged, we focus on CLIP (Contrastive Language-Image Pre-training), a large-scale model with strong zero-shot performance on downstream classification tasks (Radford et al., 2021).

**Contrastive Language-Image Pre-training (CLIP)** CLIP consists of two parallel encoders for processing images and text, whose outputs are then projected into a shared embedding space. The text encoder is a Transformer (Vaswani et al., 2017) following the architecture described in Radford et al. (2019), while the image encoder is a Vision Transformer (ViT) with a patch size of 16 (Dosovitskiy et al., 2020). For our experiments, we utilize the open-sourced pretrained CLIP models.

**Training** Because the image and text features live in the same embedding space, the cosine similarity between any embedded image and text description can be computed. CLIP uses these as prediction probabilities for classifying an image with the correct text caption (or vice versa) across batches. Formally, denote $I$ and $T$ as the set of image and text features in a single batch. The prediction probability for the $i$th image and $j$th caption in the batch is given by

$$p(\boldsymbol{T}_j \mid \boldsymbol{I}_i) = \frac{\exp\left(\cos(\boldsymbol{T}_j, \boldsymbol{I}_i)/\tau\right)}{\exp\left(\cos(\boldsymbol{T}_j, \boldsymbol{I}_i)/\tau\right) + \sum_{k \neq j} \exp\left(\cos(\boldsymbol{T}_k, \boldsymbol{I}_i)/\tau\right)}$$

| Linear Probe | Prompt Tuning | Compacter | Layer Norm | Adapter | Full Fine-tuning |
|---|---|---|---|---|---|
| 769 · (# classes) | 4096 | 19,360 | 65,536 | 301,344 | 149,620,737 |

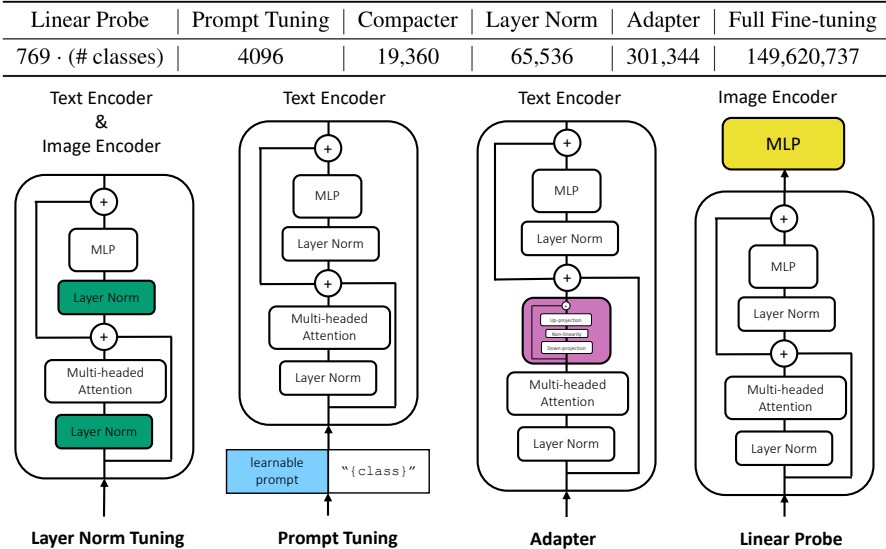

Figure 2: Parameter count and architectures of fine-tuning methods. All of the fine-tuning methods we consider tune only a small fraction of the total number of parameters and act in different ways on the model. LayerNorm-tuning only trains existing Layer Normalization parameters across all Transformer layers. The remaining approaches inject additional parameters which act on different parts of the model: the input, intermediate activations, and output. Prompt tuning prepends a learnable prompt to the input text embeddings of classes. Adapter modules are composed of a linear down-projection, non-linearity, and linear up-projection, and are inserted inside the Transformer layers of the text encoder after the attention block. Linear probe directly classifies classes from the output of the image encoder.

where $\tau$ is a learnable temperature parameter. CLIP is trained with a contrastive loss accordingly across 400 million pairs of image and text captions collected online (Radford et al., 2021).

**Inference** For a downstream classification task at test time, CLIP first embeds the textual descriptions of all classes. These descriptions may range from a phrase like "a photo of a <class>" to heavily engineered embeddings ensembled over 80 different templates (Radford et al., 2021). Each image is then classified using the embedded classes as labels and the prediction probabilities described above. Notably, this inference scheme allows CLIP to be transferred zero-shot to any downstream image classification task. Radford et al. (2021) show that zero-shot CLIP is competitive with a fully supervised ResNet (He et al., 2016) baseline on a suite of image classification tasks.

## 3 METHODOLOGY: FINE-TUNING LARGE-SCALE PRETRAINED MODEL

Although zero-shot CLIP performs well on natural images and general object classification datasets, its performance degrades quickly on more abstract tasks from out-of-distribution data. Even on a simple dataset like MNIST (LeCun, 1998), the zero-shot CLIP model (ViT-B/16) we test attains an accuracy of only 55%. Substantial gains can be achieved by fine-tuning the pre-trained model, but many such strategies have emerged across tasks in vision and language and it's unclear which to use on diverse downstream settings. For this reason, we provide an extensive study of adaptation approaches. Figure 1 illustrates the fine-tuning methods we consider in context of CLIP while Figure 2 shows more detailed information regarding each approach.

We propose a general taxonomy of fine-tuning approaches and consider three major classes: (a) methods which only fine-tune existing parameters, (b) methods which freeze existing parameters and add new parameters, and (c) methods which combine (a) and (b). We first consider two methods in (a) which only fine-tune existing parameters.

### 3.1 FINE-TUNING EXISTING PARAMETERS

**Full Model Fine-tuning** The simplest approach to fine-tuning is to train all of the model parameters on the downstream task. However, this is unstable and doesn't scale well to CLIP-size models with hundreds of millions of parameters. Our empirical results show this behavior as well.

**LayerNorm Tuning**   Instead of full model fine-tuning for large-scale models, we can tune a small subset of chosen parameters when the downstream data is scarce. In fact, Frankle et al. (2020) show that just tuning Batch Normalization (Ioffe & Szegedy, 2015) parameters from a random initialization can be highly expressive. In a similar vein, we investigate tuning the parameters of Layer Normalization (LayerNorm) layers (Ba et al., 2016). Unlike Batch Normalization, LayerNorm applies per-element normalization across mini-batches. Given a mini batch of inputs $\boldsymbol{x}$, LayerNorm transforms this as

$$\boldsymbol{y} = \frac{\boldsymbol{x} - \mathrm{E}[\boldsymbol{x}]}{\sqrt{\mathrm{Var}[\boldsymbol{x}] + \epsilon}} \cdot \gamma + \beta$$

where the mean and variance are calculated over the normalized dimensions and $\gamma, \beta$ are learned parameters. Because the image and text encoders in CLIP share the same underlying Transformer architecture, in LayerNorm Tuning, we fine-tune the Layer Normalization parameters $\gamma, \beta$ across all layers of both encoders. These parameters are 768-dimensional and 512-dimensional for the image and text encoders respectively.

### 3.2   Fine-tuning New Parameters

An alternative paradigm is to inject new parameters which can more effectively adapt to downstream tasks. These new parameters can act at various stages of a pre-trained model: on the output, input, or intermediate activations.

**Linear Probe**   The classic method of training a linear probe on top of frozen features is an example of adding new parameters which act on the model output. Given a pre-trained CLIP model, we discard the text encoder, freeze the image encoder, and learn a linear layer on top of the image features before they're projected to the shared embedding space. The linear layer maps the penultimate image features to logits from which class predictions are made. While this simple method is popular and effective, it's parameter-inefficient for tasks with higher number of classes and fails to leverage any of the language information contained in CLIP.

**Prompt Tuning**   Alternatively, we can consider adding parameters which act on the model input. Such an approach known as prompt tuning has emerged as a parameter-efficient fine-tuning method in language (Li & Liang, 2021; Lester et al., 2021). A fixed number of continuous vectors (a "prompt") is prepended to the model input and optimized throughout training. Similar to concurrent work by Zhou et al. (2021), we apply prompt tuning to image classification with CLIP. For the model input, we embed the raw text of the classes without a template and prepend a continuous prompt of fixed length. During training, the prompt is learned using a cross-entropy loss according to the prediction probabilities detailed in Section 3.1. Although prompt tuning can be applied in the same way for transformer-based visual encoders, we find that only applying it for the text encoder produces better and more stable results.

Prompt tuning is parameter-efficient and removes the need for manual prompt engineering e.g. specifying "a photo of a <class>, a type of flower" for a downstream task on flower classification. Ideally, the learned prompts would contain such domain-specific information. However, prompt tuning suffers from high variance during training and is sensitive to initialization.

**Adapter and Compacter Networks**   The above two approaches inject parameters which act either at the end of the network (linear probe) or at the beginning (prompt tuning). A third option is to inject new parameters for the downstream task within the layers of the network itself. This idea has been popularized as an efficient transfer learning method in language (Houlsby et al., 2019). For Transformer-based architectures, a common strategy is to insert a block of learnable parameters after feed forward layers or the attention mechanism.

Adapter networks insert learnable adapter blocks after the feed forward layers in each Transformer layer (Houlsby et al., 2019). Each block follows a bottleneck architecture and is composed of a linear down-projection, non-linearity, and linear up-projection as shown in Figure 2. However, for architectures with many stacked Transformer layers and larger hidden dimensions, adapter modules are parameter-inefficient.

To alleviate this issue, Mahabadi et al. (2021) introduce compacter modules which follow the same architecture but use low-rank parameters and hypercomplex multiplication to improve parameter efficiency. Specifically, if the down-projection layer maps $\boldsymbol{x} \in \mathbb{R}^m \rightarrow \boldsymbol{W}\boldsymbol{x} + \boldsymbol{b} \in \mathbb{R}^d$ where

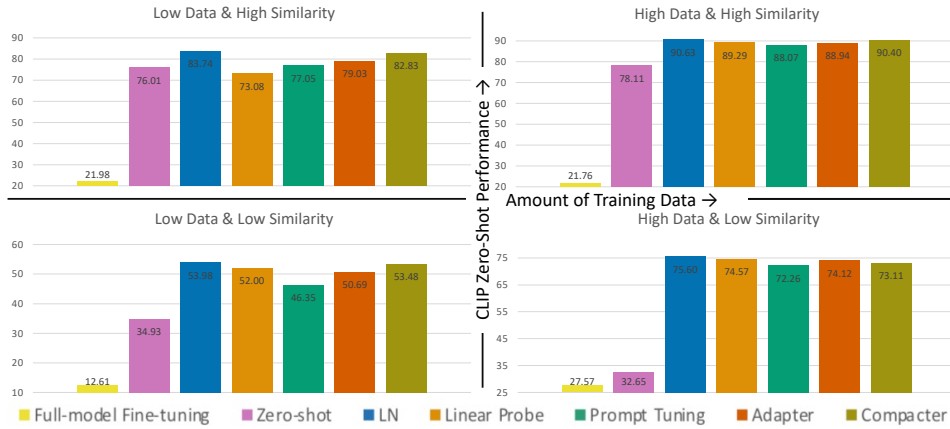

Figure 3: Comparison of fine-tuning methods across different regimes of training data and CLIP zero shot performance. Within each quadrant, results are averaged over all corresponding datasets. LayerNorm tuning is the strongest baseline and performs the best in all regimes. All fine-tuning methods generally provide a large benefit over zero-shot CLIP.

$W \in \mathbb{R}^{m \times d}, b \in \mathbb{R}^d$ are learned parameters and $d \ll m$, compacter modules represent $W$ as

$$W = \sum_{i=1}^{n} A_i \otimes \left( s_i t_i^T \right)$$

where $A_i$ are global weights shared across Transformer layers and $s_i, t_i$ are local, rank-1 weights. We insert Adapter and Compacter modules across the Transformer layers in the text encoder.

### 3.3 COMBINING LAYERNORM-TUNING WITH FINE-TUNING METHODS

While we find that just LayerNorm tuning by itself is a strong baseline, an additional benefit is that it can be combined with any other fine-tuning method given that the underlying model architecture contains Layer Normalization parameters. The parameters of the alternative method can simply be fine-tuned simultaneously with the Layer Normalization parameters. For example, Houlsby et al. (2019) combine LayerNorm tuning with Adapter modules in their Adapter network for language tasks.

**Fine-tuned LayerNorm as Initialization** We propose an additional approach for combining Layer-Norm Tuning with other fine-tuning methods. We first finetune a CLIP model using only LayerNorm tuning. The weights of this model can then be used as initialization for any arbitrary subsequent fine-tuning method. In this multi-stage process, we effectively distill the fine-tuned LayerNorm model through the LayerNorm parameters to the secondary fine-tuning method.

## 4 EXPERIMENTS

**Setup** The goal of this work is to study transfer learning to downstream vision tasks. However, the downstream transfer performance depends on two key factors: the amount of training data present as well as the distribution of that training data relative to what the model was pre-trained on. We aim to investigate transfer learning across both these dimensions. To do so, we create 4 different benchmark suites across these two factors: low data and high similarity, low data and low similarity, high data and high similarity, and high data and low similarity. Results along these axes are shown in Figure 3.

**Datasets** We select a diverse set of 12 image classification datasets. We consider a subset of 7 datasets that Radford et al. (2021) use for zero-shot CLIP evaluation: MNIST (LeCun, 1998), EuroSAT (Helber et al., 2019), CIFAR-10 (Krizhevsky et al., 2009), CIFAR-100 (Krizhevsky et al., 2009), Flowers102 (Nilsback & Zisserman, 2008), DTD (Cimpoi et al., 2014), and Food101 (Bossard et al., 2014). We then test on 3 distribution shift datasets from the WILDS benchmark (Koh et al., 2021): FMoW (Christie et al., 2018), Camelyon17 (Bandi et al., 2018), and iWildCam (Beery et al., 2021). We finally benchmark on 2 few-shot adaptation tasks: MiniImageNet (Vinyals et al., 2016) and CUB (Wah et al., 2011).

**Spectrum 1: Amount of Downstream Data** We control for the amount of downstream training data by considering $k$-shot settings where $k$ samples from each class are made available during the fine-tuning training phase. For the low-data regime, we follow the few-shot evaluation scheme described in Radford et al. (2021) and train with 1, 2, 4, 8, and 16-shots, while for the high-data regime, we train with 256 and 512-shots. For each setting, we evaluate our methods on the full tests sets and average our results across all corresponding shots and three random seeds.

**Spectrum 2: Distribution of Downstream Data** We measure the similarity of downstream data to the one that CLIP was pretrained on by measuring the performance of purely transferring CLIP zero-shot. Our 12 datasets cover a range of image classification tasks including fine-grained classification, distribution shift, and few-shot learning, and they encompass diverse domains and varying downstream distributions. We split them into high and low similarity regimes according to zero-shot accuracy using a threshold accuracy of 55%. Under this scheme, MNIST, EuroSAT, DTD, FMoW, and iWildCam fall under the low similarity regime while CIFAR10, CIFAR100, Flowers, Camelyon17, and Food101 fall under the high similarity regime. For the high data, low similarity regime and high data, high similarity regime, we exclude DTD and Flowers respectively due to a lack of data.

### 4.1 EFFECTIVENESS OF INTERMEDIATE WEIGHTS

Given this characterization of 4 downstream regimes, we evaluate the performance of 5 baseline fine-tuning methods: linear probe, prompt tuning, LayerNorm tuning, adapter networks, and compacter networks. For a fair comparison and to isolate the effect of each method, we don't simultaneously tune LayerNorm parameters in any of the other 4 fine-tuning approaches. Additionally, we compare with a sixth baseline: zero-shot CLIP using the prompt engineering detailed in Radford et al. (2021). The addition of a strong zero-shot baseline allows us to evaluate how much our fine-tuning methods help across the different settings. Full-model fine-tuning is included as a seventh baseline for a complete, fair comparison.

Our results from Figure 3 show that LayerNorm tuning is a simple but highly effective baseline across all four regimes. It performs the best in all four regimes including the most difficult quadrant: the low-data, hard zero-shot regime. Across individual datasets, Figure 4 shows that LayerNorm tuning is consistently among the best as well. This points towards the importance of cross-modal interaction when fine-tuning as LayerNorm tuning is the only method which trains parameters in both CLIP encoders. More generally, it suggests that transfer performance on downstream vision tasks can benefit from fine-tuning grounded in language.

Furthermore, the strong performance of LayerNorm tuning as well as adapter and compacter networks in the low-data regime suggests that either fine-tuning or injecting parameters among intermediate layers in a network is key

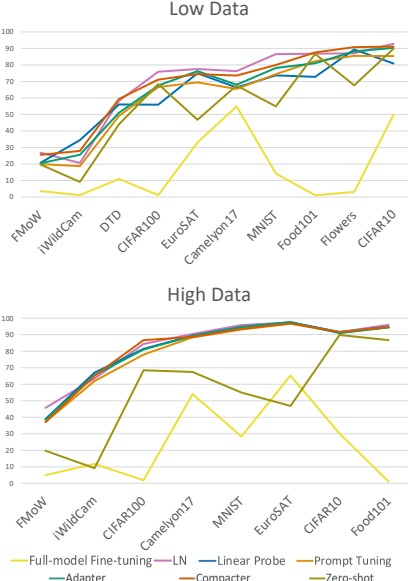

Figure 4: Accuracy of baseline fine-tuning methods across datasets in the low and high-data regimes. Datasets are ordered along the x-axis by average performance of fine-tuning methods. Flowers and Food101 datasets are ommited from the high data regime due to lack of data. We observe that LayerNorm tuning is a strong baseline across all regimes and datasets and provides substantial gains over zero-shot CLIP.

for efficient adaptation. Intuitively, acting only on the model input or output like prompt tuning or linear probe has limited expressivity compared to modifying the intermediate layers themselves, particularly in low-data regimes.

### 4.2 LEVERAGING LAYERNORM TUNING

The results of the previous section demonstrate that just LayerNorm tuning is a competitive baseline to many of the existing fine-tuning methods. We now examine how much LayerNorm tuning can benefit existing methods. We consider two ways of incorporating LayerNorm parameters: by simultaneously tuning them or by applying fine-tuned LayerNorm as initialization as described in Section 3.2. For

| Type of LN Tuning | Low-Data Regime | | | High-Data Regime | | |
|---|---|---|---|---|---|---|
| | None | Normal | As Initialization | None | Normal | As Initialization |
| Linear Probe | 62.54 | 63.55 | **66.61** | 81.93 | 83.84 | **84.36** |
| Prompt Tuning | 61.70 | 62.82 | **63.95** | 80.26 | **83.69** | 83.17 |
| Adapter | 64.82 | 66.23 | **66.63** | 81.53 | 83.31 | **83.63** |
| Compacter | 68.15 | **69.69** | 68.76 | 81.75 | **83.61** | 83.17 |

Table 1: Effect of combining LayerNorm tuning with fine-tuning methods. Normal refers to LN tuning by simultaneously tuning LayerNorm parameters with the specified fine-tuning method. For all approaches, the addition of LayerNorm tuning in either form provides a significant performance boost. Linear probe and prompt tuning receive the largest benefit when combined with any form of LayerNorm tuning.

each of the remaining fine-tuning methods (linear probe, prompt tuning, adapter networks, compacter networks), we compare the baseline method to these two variants.

We average the results over the corresponding datasets in the low and high-data regimes respectively. As Table 1 shows, incorporating any form of LayerNorm tuning increases performance compared to the normal baselines across all methods and regimes. Because the process of incorporating LayerNorm tuning is method-agnostic, we recommend this as a simple approach to improve transfer performance.

Across specific fine-tuning methods, linear probe receives the largest benefits from applying LayerNorm tuning first before finetuning the linear probe. We posit that this is the case as finetuning LayerNorm first effectively distills the information from the pre-trained text encoder to the LayerNorm parameters on the vision side. This is privileged information that a classical linear probe doesn't have access to and provides further evidence towards the benefit of leveraging both vision and text on downstream, unimodal tasks.

Finally, we observe that for adapter and compacter networks, using fine-tuned LayerNorm as initialization performs equivalently or slightly worse compared to simultaneously tuning the LayerNorm parameters across both data regimes. This suggests that LayerNorm tuning and inserting adapter and compacter modules may serve similar roles as fine-tuning mechanisms.

### 4.3 GENERAL RECIPES

From the previous two sections, we've seen that just LayerNorm tuning is a surprisingly effective baseline and applying it on top of other fine-tuning methods provides performance gains. We now investigate the question of what the best performing combination of fine-tuning methods across the four regimes are. We follow the experimental setup in Section 4 but compare the performance of our fine-tuning methods when combined with LayerNorm tuning first.

Figure 7 indicates that there is no clear best baseline across all four regimes but linear probe and compacter have the strongest performance when combined with fine-tuned LayerNorm as initialization. Across regimes, we observe similar results to Figure 3. In both low-data regimes, we find that prompt tuning generally perform worse than the remaining fine-tuning methods while compacter performs better. Across both high-data regimes, linear probe performs quite strongly. Notably, these trends hold despite the addition of our initialization scheme. We recommend these, as well as just Layer Norm tuning, as general recipes when selecting a fine-tuning method to use on a downstream task depending on the setting.

### 4.4 DISTRIBUTION SHIFT

While we test on the WILDS benchmark in our analysis above, we benchmark our methods to evaluate how robust they are to distribution shift in downstream tasks. We focus on domain generalization where the train and test distributions come from disjoint domains (Koh et al., 2021). For example, the Camelyon17 dataset contains training and testing images of tumor tissues coming from distinct hospitals. We compare our results to those on the public leaderboard which contains techniques for domain generalization while our models are simply trained according to an empirical risk minimization objective. For performance metrics, we use average accuracy for FMoW and Camelyon17, and Macro F1 for iWildCam.

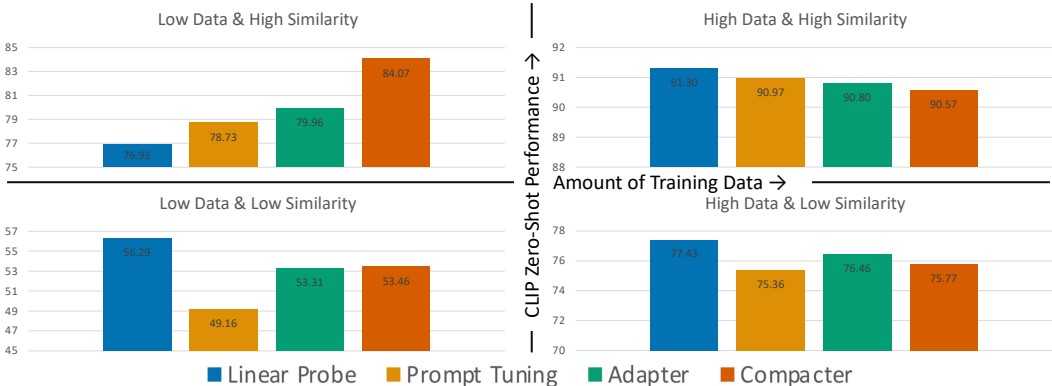

Figure 5: Comparison of fine-tuning methods initialized with fine-tuned LayerNorm across different regimes of training data and CLIP zero shot performance. Although there is no clear best combination across all four regimes, we recommend general recipes of using Linear Probe with fine-tuned initialization in the high data regimes and Compacter with fine-tuned initialization in the low data regimes.

As shown in Table 2, our methods don't quite match the best reported results, but close the gap from zero-shot CLIP significantly. We are competitive with other domain generalization specific approaches on the leaderboard despite the simplicity of our fine-tuning methods. Our results show that our fine-tuning methods are effective in adapting CLIP to difficult downstream tasks and relatively robust to factors in distribution shift.

| | FMoW | Camelyon17 | iWildCam |
|---|---|---|---|
| Zero-shot | 19.71 | 67.46 | 3.73 |
| LayerNorm Tuning | 47.59 | 90.47 | 18.52 |
| Linear Probe + LN as Initialization | 48.98 | 89.98 | 23.80 |
| Best leaderboard result | **55.5** | **91.6** | **38.5** |

Table 2: Results on the WILDS benchmark. We benchmark our fine-tuning methods on three image classification datasets and find that our fine-tuning methods improve upon zero-shot CLIP performance significantly.

## 4.5 FEW-SHOT LEARNING

Our previous experiments show results in the low-data regime, but we also apply our fine-tuning methods with CLIP to more standard meta few-shot tasks to evaluate class generalization. In particular, we consider the setting where given a labeled dataset of base classes, the objective is to identify novel classes only using a few samples. Formally, we are given a large dataset of $B$ base classes. At test time, in a single episode of a $N$-way $K$-shot few-shot task, we are given a support set with $N$ test classes and $K$ samples per class as well as a query set with $N$ test classes and $Q$ samples per class. We measure the accuracy of classifying the $N \times Q$ query images into $N$ classes (Chen et al., 2020).

We adopt our fine-tuning methods to this setting in two stages. First, we pre-train on the full dataset of the base classes in the same way we fine-tune to any of the prior downstream tasks. Then, we use the image encoder of the fine-tuned model within the Classifier Baseline proposed by Chen et al. (2020). For a given few-shot task, we compute a representative for each of the $N$ classes by averaging the embeddings of the $K$ support examples. We classify each of the $N \times Q$ query-set examples according to the cosine similarity of their embeddings to the representatives.

Using this approach, we test our fine-tuning methods on MiniImageNet and CUB in the 5-way 1-shot and 5-way 5-shot settings averaged over 600 episodes. Table 4 shows that simply applying zero-shot CLIP in this way does remarkably well, outperforming the best reported results on the public leaderboards for these tasks.

All of our fine-tuning methods provide further improvement with Linear Probe and LayerNorm tuning reaching 6% higher accuracy than the current SOTA on 5-way 1-shot MiniImageNet as shown in Tables 3 and 4. The strong performance of linear probe in this setting is expected as the Classifier Baseline doesn't utilize the CLIP text encoder at all. Of the fine-tuning methods we evaluate, Linear Probe with LayerNorm tuning is the only method which trains parameters solely in the image encoder, so its visual representations transfer the best.

| | Mini-ImageNet (1-shot) | Mini-ImageNet (5-shot) |
|---|---|---|
| Zero-shot | 86.20 | 96.56 |
| LayerNorm | 89.24 | 96.46 |
| Prompt Tuning + LN | 89.61 | 97.05 |
| Adapter + LN | 91.17 | 97.39 |
| Linear Probe + LN | **92.08** | **97.94** |
| Best leaderboard result | 82.99 | 91.50 |

Table 3: Few-shot classification accuracy on Mini-ImageNet. Just zero-shot CLIP performs strongly on few-shot adaptation, and our fine-tuning methods provide additional performance gains. A combination of Linear Probe with LayerNorm tuning performs the best, exceeding the current reported SOTA on Mini-ImageNet.

| | CUB (1-shot) | CUB (5-shot) |
|---|---|---|
| Zero-shot | 87.04 | 97.28 |
| LayerNorm | 91.40 | 98.16 |
| Adapter + LN as Initialization | 91.73 | 98.20 |
| Prompt Tuning + LN as Initialization | 92.21 | 98.20 |
| Linear Probe + LN as Initialization | 93.73 | **98.50** |
| Best leaderboard result | **94.73** | 96.28 |

Table 4: Few-shot classification accuracy on CUB. Similar to our results on Mini-ImageNet, we see that zero-shot CLIP performs strongly but fine-tuning with LayerNorm on top can produce significant improvements in accuracy. Linear Probe combined with LayerNorm performs the best again, exceeding the current SOTA in the 5-shot setting.

## 5 RELATED WORK

**Large-Scale Transformer-Based Models:** Unsupervised pre-training for language typically takes advantage of the sequential nature of text through a self-supervised prediction task. Initially, recurrent neural networks (RNNs) (Hochreiter & Schmidhuber, 1997) were the predominant deep learning architectures for unsupervised language learning and were particularly successful for machine translation (Sutskever et al., 2014). In the context of language modeling, RNNs were superseded by attention architectures (Bahdanau et al., 2015) and specifically masked self-attention Transformer architectures (Vaswani et al., 2017; Devlin et al., 2018; Radford et al., 2019). Over the last few years, Transformers have produced impressive generalization results through unsupervised pre-training of increasingly larger models on larger datasets (Brown et al., 2020).

**Multimodal Learning:** Multimodal models (Ngiam et al., 2011) are trained through tasks that leverage multiple data modalities simultaneously, such as vision and language. Examples include text to image synthesis (Reed et al., 2016) and text descriptions of visual inputs (Krishna et al., 2017). Recently, multi-modal architectures have been combined with large-scale unsupervised pre-training to achieve impressive text-to-image generation (Ramesh et al., 2021) as well as learning joint image and language embeddings (Radford et al., 2021). In particular, this work investigates how do adapt a pre-trained CLIP model, which uses noise contrastive estimation and transformers to maximize similarity between images and their text captions, to downstream tasks.

**Finetuning Pre-trained Models:** With the emergence of large-scale pre-trained vision and language models, it's becoming increasingly important to adapt such models efficiently to downstream tasks. For transfer learning in vision, the most common approach is to use a linear probe on top of pretrained image features while for language, a variety of fine-tuning approaches have emerged including variations of prompt tuning (Liu et al., 2021; Li & Liang, 2021; Lester et al., 2021; Zhou et al., 2021), adapter and compacter networks (Houlsby et al., 2019; Mahabadi et al., 2021), and multimodal approaches applicable to vision and language (Tsimpoukelli et al., 2021; Shen et al., 2021). We hope our analysis provides insight into some of these methods.

## 6 DISCUSSION: GENERAL GUIDELINES

Our work analyzes relevant questions in transfer learning of large-scale pretrained vision-and-language model to several downstream classification tasks. We evaluate 5 different fine-tuning baseline methods across 12 total image classification datasets and find that just tuning Layer Normalization parameters is a surprisingly effective, parameter-efficient baseline, and propose an effective approach to combine it with other finetuning methods.

We analyze our best-performing fine-tuning methods over different settings to find general guidelines. For all of our methods, we combine them with fine-tuned LayerNorm as initialization. For the low-data regime, we recommend using fine-tuning approaches which inject or modify intermediate parameters like LayerNorm tuning, Adapter networks, and Compacter networks. For the high-data regime, we recommend using linear probe or prompt tuning. For generic settings, we recommend simply LayerNorm tuning. Code to reproduce the experiments will be made available. We hope that this work will lead to a broader future research in efficient adaptation of large-scale pre-trained models, not just limited to vision-and-language models.

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

# A APPENDIX

## A.1 DATASET INFORMATION

| Dataset | Classes | Train | Validation | Test |
|---------|---------|-------|------------|------|
| MNIST | 10 | 58,000 | 2,000 | 10,000 |
| EuroSAT | 10 | 15,000 | 2,000 | 10,000 |
| CIFAR-10 | 10 | 48,000 | 2,000 | 10,000 |
| CIFAR-100 | 100 | 48,000 | 2,000 | 10,000 |
| DTD | 47 | 1,880 | 1,880 | 1,880 |
| Flowers102 | 102 | 1,632 | 408 | 6,149 |
| Food-101 | 101 | 60,600 | 15,150 | 25,250 |
| FMoW | 62 | 76,863 | 19,915 | 22,108 |
| iWildCam | 182 | 129,809 | 14,961 | 42,791 |
| Camelyon17 | 2 | 302,436 | 34,904 | 85,054 |
| CUB | 200 | 5,891 | 2,932 | 2,965 |
| MiniImageNet | 100 | 38,400 | 9,600 | 12,000 |

Table 5: Dataset information.

Table 5 shows information about the number of classes and the train, validation, and test split for each downstream dataset we evaluate. For MiniImageNet, the 100 classes are split into 64, 16, and 20 classes for training, validation, and testing respectively. For CUB, the 200 classes are split into 100, 50, and 50 classes for training, validation, and testing respectively.

## A.2 CLIP IMPLEMENTATION

We implement our adaptation methods on top of the PyTorch implementation provided by the authors of CLIP at https://github.com/openai/CLIP. For the image encoder, we use a Vision Transformer with patch size 16 (ViT-B/16) and initialize all of our models with the corresponding pre-trained weights. For zero-shot CLIP evaluation, we perform the prompt engineering described in Radford et al. (2021) and ensemble the features from the text encoder across 80 templates [2].

## A.3 TRAINING DETAILS

**Input Processing** For the raw images, we preprocess them with normalization, resizing, and random cropping to size 224 by 224. For the textual descriptions, we encode them using a byte pair encoding like Radford et al. (2021).

**Hyperparameters of fine-tuning methods** For prompt tuning, we use a prompt of 8 512-dimensional vectors. For adapter networks, we use a bottleneck dimension of size 24. For compacter networks, we use the same bottleneck dimension and have 4 global weights of dimension 4 by 4.

**Training Procedure** For all fine-tuning methods, we train with a cross-entropy loss. We use the AdamW optimizer with an initial learning rate of 5e-4 and weight decay of 0.02 as well as a cosine annealing scheduler. We evaluate our model on the validation set after every epoch and keep the best-performing checkpoint. In the low-data regime, we train for 100 epochs and use a validation set of size # of shots except for Flowers where we use $\min\{\text{\# of shots}, 4\}$. In the high-data regime, we train for 50 epochs and use a validation set of size $(\text{\# of shots})/10$. Due to data constraints, for CIFAR-100, we train with 256 and 450-shots (instead of 256 and 512-shots) in the high-data regime. For CUB and MiniImageNet, we use the same hyperparameters and train for 100 epochs on the base classes.

---

[2] https://github.com/openai/CLIP/blob/main/notebooks/Prompt_Engineering_for_ImageNet.ipynb

