# OpenReview forum: "How to Adapt Your Large-Scale Vision-and-Language Model"
_ICLR.cc/2022/Conference — ICLR 2022 Submitted_

### Official Review · Reviewer_Dgds · 2021-10-23

**Correctness:** 3
**Technical Novelty And Significance:** 2
**Empirical Novelty And Significance:** 2
**Recommendation:** 5
**Confidence:** 4

**Main Review:**

- Strengths
    - The proposed LN tuning is simple and seems quite strong compared to other more complex methods.

- Weaknesses:
    - While the authors claim that the proposed method is useful for vision and language models, only a single model (CLIP) is used for these studies. Though the method design is almost model-agnostic, it is not clear whether this approach generalizes to other models.
    - The tasks studied in this work are all image classification tasks, there are no tasks that are normally recognized as vision-and-language tasks, such as text-image retrieval or image captioning [1]. If the authors' intention is to use this method for image classification tasks, it would be better to change the title a bit as "How to adapt your large-scale pre-trained models for visual representations learning", instead of the current confusing one of "vision-and-language model".
    - In the high-data regime, it seems also feasible to finetune all the model parameters, how does this result compare to the considered baselines in this work?
    - The best performance is often achieved by first finetuning the LN parameters, then using another approach like Adapter for a 2nd stage finetuning. This two-stage finetuning process could be cumbersome.
    - After LN finetuning, it seems that additionally finetune with many of the other methods actually hurts performance. For example, under low data & high similarity setting, with only LN finetuning, the performance is 83.74 (see Fig. 3, top-left), while the performance after further finetuned with other methods generally falls (see Fig. 4, top-left), only Compactor shows a minimal gain with a performance of 84.07. This conclusion seems also applicable to other settings in these two figures. This seems to suggest that we should probably only use LN.

- Miscs:
    - Some of the references are not linked, e.g., the 3rd paragraph in Sec 4.5, "Table 4" is not hyper-linked
    - "Normal" setting in Table 1 is not clearly explained. I made an educated guess as "simultaneously tuning them" when evaluating the work.

[1] Chen et al. Microsoft coco captions: Data collection and evaluation server. *arXiv 2015*

**Summary Of The Paper:**

This paper proposed a new method (LayerNorm tunning) for finetuning a pre-trained vision-and-language model. While simple, it is shown this method works competitively with other methods (e.g., prompt tuning) in various settings. For the low data & high similarity setting (Fig 3), it shows the best performance across the methods. The authors further claimed LN could be used to boost the performance of general finetuning methods like linear probe or prompt tuning. However, this seems not useful as the performance typically drops when compared to the LN baseline. In addition, the proposed methods are only evaluated in a single model, it is not clear whether they are generalizable.

**Summary Of The Review:**

The paper proposed a useful technique LN tunning for fine-tuning the pre-training CLIP model for downstream image classification tasks. However, the generalizability of this approach is not demonstrated, and the proposed recipe of combing LN tunning with other fine-tuning methods seems also not useful as it typically shows even lower performance than only using LN tuning. Overall, this paper does not provide enough support for it to be published at ICLR.

---

> ### Author Response · Authors · 2021-11-22
> **Author Response: new experiments and addressing other concerns**
>
> We thank the reviewer for their valuable feedback and hope that we have addressed your concerns below. Please let us know whether there are any remaining points – which cause you to continue recommending your current score – that we can address. Thank you!
>
> > *"Full-model fine-tuning results"*
> - We find that generally across both the low and high data settings, full-model fine-tuning is too unstable and catastrophically fails the majority of the time compared to the analyzed baselines. This is because CLIP has around 150M parameters and originally needed 400M examples during training. Compared to this even the downstream task which we are calling *high data* in relative sense has at most 100K examples, hence, fine-tuning the full model is not an oracle in this setting.
> - In the absolute best case, where we are able to fine-tune the full model, it results in performance slightly worse than zero-shot CLIP. We attribute this behavior to the difficulty of optimizing all of the parameters in CLIP and improving upon its initialization. Notably, the parameter efficient methods we examine and propose don’t suffer from this problem.
> - Full results can be seen here: https://drive.google.com/file/d/1evCSx7Aqg9S1QS3oMGuaIIX47n63hy_X/view?usp=sharing
>
> > *"Though the method design is almost model-agnostic, it is not clear whether this approach generalizes to other models."*
> - We focus on CLIP in our analysis as it is one of the few open-sourced models which has been pre-trained on large-scale unfiltered data from the internet. Furthermore, similar large-scale vision-and-language models have been developed. For example, a similar large-scale model known as ALIGN [Jia etal., 2021] was published earlier this year and a scaled-up version of CLIP called BASIC [Pham etal., 2021] was published just last week. Our analysis should apply to these models as the difference between them is primarily the amount of data and parameters.
> - We also believe our results are of interest to many researchers because since its release earlier this year, CLIP has been used across robotics [CLIPort: Shridhar etal., CoRL 2021], videos [CLIP-It!: Narasimhan etal, 2021], image synthesis [StyleGAN-NADA: Gal etal 2021], and more.
>
> > *”The tasks studied in this work are all image classification tasks…it would be better to change the title a bit as "How to adapt your large-scale pre-trained models for visual representations learning"”*
> -  We agree with this concern. Because our work focus on downstream classification tasks, we will combine your title suggestion with Reviewer # Vfao who suggested scoping it by adding “to Downstream Image Classification Tasks”. We feel that is more accurate/stricter than saying visual representation learning.
> - Hence, **we will modify our title to** "How to Adapt Your Large-Scale Pre-trained Model for Downstream Image Classification". Hope that works for you as well as it makes the scope narrower than your suggestion.
>
> > *”The best performance is often achieved by first finetuning the LN parameters, then using another approach like Adapter for a 2nd stage finetuning. This two-stage finetuning process could be cumbersome.”*
> - The two-stage fine-tuning process required for “fine-tuned LN as initialization” only involves training two CLIP-style models sequentially: the first is trained by only fine-tuning the LayerNorm parameters while the second is trained with the first as initialization. The overall method is still simple to use and no more complex than just using the fine-tuning method if the LayerNorm tuned models have already been trained.
>
> > *”After LN finetuning, it seems that additionally finetune with many of the other methods actually hurts performance. This conclusion seems also applicable to other settings in these two figures. This seems to suggest that we should probably only use LN.”*
> - Comparing Figures 3 and 4, in the low data regime, we note that just LayerNorm tuning is a very strong baseline. However, in the high data regime in Figure 4, combining the fine-tuning methods with LayerNorm tuning performs equally as well or better. Moreover, the few-shot classification results for MiniImageNet and CUB in Tables 3 and 4 demonstrate that combining LayerNorm tuning with a linear probe can push the performance past the current SOTA. Ultimately, our conclusions regarding LayerNorm tuning are two-fold: it serves as a strong baseline and initial fine-tuning method alone, but can also benefit existing fine-tuning methods in different settings.
>
> > *"Normal" setting in Table 1 is not clearly explained. I made an educated guess as "simultaneously tuning them" when evaluating the work.*
> - We apologize for the lack of clarity regarding the “Normal” setting. We indeed use the term to refer to simultaneously tuning the parameters of the specified fine-tuning method and LayerNorm layers. We have updated the paper to clarify this.

---

### Official Review · Reviewer_Vfao · 2021-11-01

**Correctness:** 3
**Technical Novelty And Significance:** 2
**Empirical Novelty And Significance:** 3
**Recommendation:** 5
**Confidence:** 4

**Main Review:**

Strengths:

1. The LayerNorm-tuning for adapting CLIP to downstream tasks is shown to be effective.

2. A simple yet effective scheme that combines LayerNorm-tuning with other fine-tuning methods is proposed to obtain competitive performance across the board.

3. A thorough comparison of different adaptation methods is provided in four scenarios across two spectra.

Weaknesses:

1. The paper title “How to Adapt Your Large-Scale Vision-and-Language Model” is a bit of over-claimed. It should be toned down to “How to Adapt CLIP to Downstream Image Classification Tasks”.

2. The technical novelty of this paper seems very limited. The three methods for fine-tuning new parameters have been extensively studied in existing works. When coming to adapting CLIP, there should be something different. That is, novel fine-tuning methods are needed in adapting CLIP.

3. The experimental evaluation is far from sufficient. (1) For “LayerNorm-Tuning”, the parameters of all LayerNorm layers are set to be learnable. I wonder if the results could be better when only the last K LayerNorm layers are updated (e.g., K=5). (2) For “Full Model Fine-tuning”, it should also be evaluated in the experiments by only updating the parameters of the last layers of the model. (3) I wonder if the proposed method is effective in other downstream tasks other than image classification.


**Summary Of The Paper:**

This paper has extensively studied how to adapt the large-scale pre-trained vision-language model CLIP for downstream tasks. Several fine-tuning methods are analyzed across a diverse set of image classification tasks along two spectra. Further, a simple yet effective strategy that combines LayerNorm-tuning with general fine-tuning methods is proposed to improve their performance and benchmark them on few-shot classification tasks.

**Summary Of The Review:**

This paper had been well written if entitled “How to Adapt CLIP to Downstream Image Classification Tasks”. However, considering the insufficient experimental evaluation and limited technical novelty, I could only give it a score of 5.

---

> ### Author Response · Authors · 2021-11-22
> **Author Response: title update and new experiments**
>
> We thank the reviewer for their valuable feedback and hope that we have addressed your concerns below. Please let us know whether there are any remaining points – which cause you to continue recommending your current score – that we can address. Thank you!
>
> > *"The paper title... should be toned down to “How to Adapt CLIP to Downstream Image Classification Tasks”."*
> -  We agree with this concern. Although we use CLIP specifically, large-scale models with similar architectures are becoming increasingly popular. For example, a similar large-scale model known as ALIGN [Jia etal., 2021] was published earlier this year and a scaled-up version of CLIP called BASIC [Pham et al., 2021] was published just last week. Our analysis should apply to these models as the difference between them is primarily the amount of data and parameters. In addition, Reviewer # Dgds also suggested title change to: “How to adapt your large-scale pre-trained models for visual representations learning”.
> - Hence, **we will combine your suggestion with Reviewer # Dgds** to scope it to image classification: "How to Adapt Your Large-Scale Pre-trained Model for Downstream Image Classification". Please let us know if you will be satisfied with this change.
>
> > *"Full-model fine-tuning results"*
> - Reviewers requested full-model fine-tuning results to compare to the analyzed baselines and to provide a fair perspective. We find that generally across both the low and high data settings, full-model fine-tuning is too unstable and catastrophically fails the majority of the time compared to the analyzed baselines. This is because CLIP has around 150M parameters and originally needed 400M examples during training. Compared to this even the downstream task which we are calling *high data* has at most 100K examples, hence, fine-tuning the full model is not an oracle in this setting.
> - In the absolute best case, where we are able to fine-tune the full model, it results in performance slightly worse than zero-shot CLIP. We attribute this behavior to the difficulty of optimizing all of the parameters in CLIP and improving upon its initialization. Notably, the parameter efficient methods we examine and propose don’t suffer from this problem.
> - Full results can be seen here: https://drive.google.com/file/d/1evCSx7Aqg9S1QS3oMGuaIIX47n63hy_X/view?usp=sharing
>
> > *Ablation of LayerNorm parameters during fine-tuning*
> - As requested by the reviewer, we provide an ablation of LayerNorm tuning where we consider only updating the LayerNorm parameters in the last K residual blocks of the vision and text transformers for K = [2, 4, 6, 8, 10, 12]. We consistently find that across all datasets and shots, increasing the number of residual blocks with tuned LayerNorm parameters results in improved performance up to K = 12. This indicates that all tuned LayerNorm parameters contribute to the final output representation — this is consistent with our proposed approach.
> - The full results are shown here: https://drive.google.com/file/d/1YEZJruYsGH2DISIDCwhraArwl3EKCuwH/view?usp=sharing
>
> > *”The technical novelty of this paper seems very limited. The three methods for fine-tuning new parameters have been extensively studied in existing works. When coming to adapting CLIP, there should be something different.”*
>
> We address novelty and contribution concerns of several reviewers as follows:
>
> - **Adaptation Methods (NLP vs Vision)**: The adaptation methods which we study have *previously been studied predominantly in NLP settings for language models* while we study them for classification tasks *in vision*. We believe this is an important contribution as large-scale pre-trained models like CLIP have become popular but are not easy to reproduce or retrain given their ultra large scale (150M params, 400M examples) for every downstream task. Furthermore, it’s unclear which fine-tuning methods to use in different situations.
>
> - **Results and Conclusions**: We demonstrate several new conclusions using our analysis of LayerNorm tuning. First, in a vacuum, simple LayerNorm tuning is a strong yet simple baseline (and relatively model and task-agnostic) across a diverse set of downstream tasks. Second, combining it with other methods like linear probe or compactor networks can produce large gains over the baseline methods. We demonstrate this most clearly in the first two rows of Table 1 and our best performing results in Tables 3 and 4. Third, we note that the strong performance of linear probe with fine-tuned LN as initialization provides evidence towards the strength of LayerNorm tuning and the importance of joint vision-and-language training for strong visual representations.
>
> - **CLIP as base model**: We choose CLIP for this analysis as it is one of a kind in scale and format among open-sourced models, and had an enormous impact across robotics [CLIPort: Shridhar etal., CoRL 2021], videos [CLIP-It!: Narasimhan etal, 2021], image synthesis [StyleGAN-NADA: Gal etal 2021], and much more.

---

### Official Review · Reviewer_aQ8A · 2021-11-02

**Correctness:** 3
**Technical Novelty And Significance:** 2
**Empirical Novelty And Significance:** 2
**Recommendation:** 5
**Confidence:** 3

**Main Review:**


Pro:

It is good to know that LayerNorm tuning is quite effective for CLIP and combining it with other approaches gives even better performance.


Con:

1. Limited novelty.

The methods used in this paper are existing methods and the authors do a simple combination and benchmark them with CLIP on various datasets.

The conclusion I could draw is 1) layer norm tuning works well and 2) one can combine it with other methods (essentially have more parameters to tune) to achieve better results. I am not sure if these conclusions are that exciting and sufficient for an accepted paper.


2. Missing full-model fine-tuning.

The authors argue full-model fine-tuning as "inefficient" and exclude it from comparison. However, I think full-model fine-tuning results are important to include.

a. It is important to see where the upper bound is and results of full-model fine-tuning would put the current results in perspective.

b. The authors need to be more clear about what "inefficient" means. Does it mean fine-tuning time cost? If it refers to the fine-tuning time cost, it is affected by many factors and is not just dependent on parameter count. And under the setting in this paper, I am not sure how "inefficient" full fine-tuning is, compared to the shown approaches.
   For example, for prompt tuning, even if most parameters are not trainable, gradients still need to be calculated for back-propagation to the input. Thus, the forward/backward computational cost should be the same for prompt tuning and full model fine-tuning. This is similar to layer-norm tuning (depending on the location of the first layer norm layer).
   The efficiency of prompt tuning over full-model fine-tuning mostly comes from 1) less optimizer overhead and 2) less communication cost of parameters during multi-GPU training. However, CLIP is not a significantly large model by today's standards (the smallest CLIP is ResNet 50) and on the datasets we are adapting to, we often do not need many GPUs to fine-tune the model. I have to suspect that the bottleneck is not on either the optimizer overhead or communication cost when we fine-tune CLIP on those small datasets (please correct me if my assumption is wrong). Thus I am curious to see the actual speedup prompt tuning holds over full fine-tuning.




**Summary Of The Paper:**

The paper investigates a range of techniques for adapting CLIP to different tasks. They find that only tuning the LayerNorm is effective and further combining it with other adapting approaches delivers better performance.


**Summary Of The Review:**

The paper presents an empirical study of how to fine-tune CLIP for downstream tasks.

My main concerns are: 1) from my subjective view, I am not sure the conclusions are significant enough; 2) full fine-tuning results are missing.

I recommend adding the full fine-tuning results for a full picture and being more specific about the reason to exclude full fine-tuning. It would be good to provide a compelling reason to favor these methods over full fine-tuning in practice.

---

> ### Author Response · Authors · 2021-11-22
> **Author Response: new experiments and addressing other concerns**
>
> We thank the reviewer for their valuable feedback and hope that we have addressed your concerns below. Please let us know whether there are any remaining points – which cause you to continue recommending your current score – that we can address. Thank you!
>
> > *"Full-model fine-tuning results"*
> - Multiple reviewers requested full-model fine-tuning results to compare to the analyzed baselines and to provide a fair perspective. We find that generally across both the low and high data settings, full-model fine-tuning is too unstable and catastrophically fails the majority of the time compared to the analyzed baselines. This is because CLIP has around 150M parameters and originally needed 400M examples during training. Compared to this even the downstream task which we are calling *high data* in relative sense has at most 100K examples, hence, fine-tuning the full model is not an oracle in this setting.
> - In the absolute best case, where we are able to fine-tune the full model, it results in performance slightly worse than zero-shot CLIP. We attribute this behavior to the difficulty of optimizing all of the parameters in CLIP and improving upon its initialization. Notably, the parameter efficient methods we examine and propose don’t suffer from this problem.
> - Full results can be seen here: https://drive.google.com/file/d/1evCSx7Aqg9S1QS3oMGuaIIX47n63hy_X/view?usp=sharing
>
> > *”The authors need to be more clear about what "inefficient" means… For example, for prompt tuning, even if most parameters are not trainable, gradients still need to be calculated for back-propagation to the input”*
> - We apologize for the terminology confusion; we agree that the forward and backward computational cost for prompt tuning and full model fine-tuning are similar due to the calculated gradients. We mean to say that the training of full-model fine-tuning is significantly more unstable than that of fine-tuning methods like prompt tuning – as shown in the previous answer. In particular, full-model fine-tuning requires simultaneously optimizing nearly 150 million parameters while prompt tuning only requires optimizing 4096. We have edited the manuscript to clarify this point.
>
> > *Limited Novelty: ”The methods used in this paper are existing methods and the authors do a simple combination and benchmark them with CLIP on various datasets. I am not sure if these conclusions are that exciting and sufficient for an accepted paper.”*
>
> Below we address reviewer's novelty concern grouped in three categories:
>
> - **Adaptation Methods (NLP vs Vision)**: The adaptation methods which we study have *previously been studied predominantly in NLP settings for language models* while we study them for classification tasks *in vision*. We believe this is an important contribution as large-scale pre-trained models like CLIP have become popular but are not easy to reproduce or retrain given their ultra large scale (150M params, 400M examples) for every downstream task. Furthermore, it’s unclear which fine-tuning methods to use in different situations.
>
> - **CLIP as base model**: We choose CLIP for this analysis as it is one of a kind in scale and format among open-sourced models. Reviewer aQ8A raises the question of “whether our conclusions are that exciting and sufficient for an accepted paper". We believe they are of interest to many researchers because of the enormous impact of CLIP. Since its release earlier this year, it has been used across robotics [CLIPort: Shridhar etal., CoRL 2021], videos [CLIP-It!: Narasimhan etal, 2021], image synthesis [StyleGAN-NADA: Gal etal 2021], and more.
>
> - **Results and Conclusions**: We demonstrate several new conclusions using our analysis of LayerNorm tuning. First, in a vacuum, simple LayerNorm tuning is a strong yet simple baseline (and relatively model and task-agnostic) across a diverse set of downstream tasks. Second, combining it with other methods like linear probe or compactor networks can produce large gains over the baseline methods. We demonstrate this most clearly in the first two rows of Table 1 and our best performing results in Tables 3 and 4. Third, we note that the strong performance of linear probe with fine-tuned LN as initialization provides evidence towards the strength of LayerNorm tuning and the importance of joint vision-and-language training for strong visual representations.

---

### Official Review · Reviewer_GtTP · 2021-11-08

**Correctness:** 2
**Technical Novelty And Significance:** 1
**Empirical Novelty And Significance:** 1
**Recommendation:** 3
**Confidence:** 4

**Main Review:**

Strengths
* I like the motivation of the paper -- the vision community broadly uses linear probe as the standard protocol for transfer learning, yet the prominence of prompt learning in NLP clearly shows that there can be better methods to transfer knowledge of large-scale pre-trained models in the vision domain.

Weaknesses
* In Figure 3-4, it seems that LayerNorm performs better with an extremely small margin, which could easily be reversed with initialization or hyperparameter search for each fine-tuning method. This makes me question why the paper is particularly focused on analyzing LayerNorm in the second half of the paper. Also, combining LayerNorm provides minimal improvement.
* Instead of averaging and phrasing as "low data" vs. "high data" in the paper, it would be more accurate to plot the actual 1-512 shots on the graph. Averaging it makes it hard to grasp the correct trend of each fine-tuning method over the dataset scale.
* Furthermore, it seems extremely arbitrary to divide the distribution of downstream data as "high and low similarity" with an arbitrary threshold of 55%.


**Summary Of The Paper:**

The goal of the paper is to investigate how to efficiently adapt large-scale pre-trained vision-language models (e.g. CLIP) to downstream tasks. Their paper is based on the observation that while the vision community dominantly uses linear probe as the standard protocol, other approaches such as prompt learning are utilized in language. They compare and analyze several fine-tuning methods -- linear probe, prompt tuning, adapter and compacter networks -- across 12 downstream classification tasks. Focusing on LayerNorm, they further demonstrate combining LayerNorm tuning with existing fine-tuning methods to improve performance.


**Summary Of The Review:**

Overall, this paper seems to have limited contributions for adapting pre-trained vision-language models to downstream tasks. The depth of the analyses is limited and does not provide interesting and novel findings to the community.

---

> ### Author Response · Authors · 2021-11-22
> **Author response: new experiments and ablations**
>
> We thank the reviewer for their valuable feedback and hope that we have addressed your concerns below. Please let us know whether there are any remaining points – which cause you to continue recommending your current score – that we can address. Thank you!
>
> > *"In Figure 3-4, it seems that LayerNorm performs better with an extremely small margin, which could easily be reversed with initialization or hyperparameter search for each fine-tuning method. This makes me question why the paper is particularly focused on analyzing LayerNorm in the second half of the paper. Also, combining LayerNorm provides minimal improvement."*
> - In Figures 3 and 4, we demonstrate that LayerNorm tuning alone is a strong baseline despite the results in each quadrant of Figure 3 being averaged over multiple datasets, shots, and seeds, LayerNorm tuning consistently yields the best results over existing baseline methods in Figure 3. Across individual datasets, Figure 4 shows that LayerNorm tuning is competitive, especially in the low-data regime where it performs near the top.
> - We focus on analyzing LayerNorm in the second half of the paper because of its generality and ease of integration with other fine-tuning methods. Since the LayerNorm parameters are part of the base CLIP model, they can always be tuned alongside additional parameters that other fine-tuning algorithms add. Moreover, they can be tuned in any model, not just CLIP, which uses Layer Normalization.
> - Our results show that generally combining methods with LayerNorm tuning provide non-trivial improvement. Table 1 shows that for each baseline method, combining it with some form of LayerNorm tuning provides a gain (up to 4% for linear probe). These improvements are averaged over multiple datasets and seeds. The few-shot classification results for MiniImageNet and CUB in Tables 3 and 4 also demonstrate that combining LayerNorm tuning with a linear probe can push the performance past the current SOTA.
> - Ultimately, our conclusions regarding LayerNorm tuning are two-fold: it serves as a strong baseline and initial fine-tuning method but can also benefit existing fine-tuning methods in different settings.
>
> > *"Instead of averaging and phrasing as "low data" vs. "high data" in the paper, it would be more accurate to plot the actual 1-512 shots on the graph. Averaging it makes it hard to grasp the correct trend of each fine-tuning method over the dataset scale."*
> - We introduce the phrasing of “low data” vs “high data” to compare and analyze the performance of fine-tuning methods across different general regimes of data. We show the results of individual shots for all methods across datasets at https://drive.google.com/file/d/1evCSx7Aqg9S1QS3oMGuaIIX47n63hy_X/view?usp=sharing
> - Across individual datasets, the fine-tuning methods we test apart from full-model fine-tuning exhibit increasing performance as a function of dataset scale. In the low-data regime, this trend appears to be roughly linear for most fine-tuning methods.
> - Upon multiple reviewers’ request, we also added full-model fine-tuning result in [the same link as above](https://drive.google.com/file/d/1evCSx7Aqg9S1QS3oMGuaIIX47n63hy_X/view?usp=sharing). We find that generally across both the low and high data settings, full-model finetuning is too unstable and catastrophically fails the majority of the time compared to the analyzed baselines. This is because CLIP has around 150M parameters and originally needed 400M examples during training. Compared to this, even our high-data downstream tasks have at most 100K examples.
> - In the absolute best case, where we are able to fine-tune the full model, it results in performance slightly worse than zero-shot CLIP. We attribute this behavior to the difficulty of optimizing all of the parameters in CLIP and improving upon its initialization. Note that the parameter efficient methods we examine and propose don’t suffer from this problem.
>
> > *"...arbitrary to divide the distribution of downstream data as "high and low similarity" with an arbitrary threshold of 55%."*
> - We acknowledge that zero-shot performance of 55% is a somewhat arbitrary threshold, however, we choose to divide the distribution of downstream data in this way to compare the performance of fine-tuning methods on more in-distribution/out-of-distribution data relative to what CLIP was pre-trained on. We found this to be a natural splitting point as zero-shot CLIP scored an accuracy of at least 66% on all “high similarity” datasets. Qualitatively, we also note that the “low similarity” datasets like MNIST, EuroSAT, or DTD contain images from more unnatural, specific sources. Finally, we note that our major empirical results hold true across both regimes.
>
> > *"Overall, this paper seems to have limited contributions… interesting and novel findings to the community."*
> - We address this answer in the shared response to all above: [click here.](https://openreview.net/forum?id=EhwEUb2ynIa&noteId=E93DBZvkRRR)

---

### Author Response · Authors · 2021-11-22
**[To All] Summary of updates: new experiments and ablations**

We thank all the reviewers. To summarize, we present a thorough empirical analysis of several different ways to adapt CLIP (large-scale, pre-trained vision-and-language model) to downstream tasks. The reason for focusing on *adaptation* is because such models are too big and very expensive to be re-trained or fully fine-tuned on downstream tasks that the user may be interested in. We perform this analysis across variations in both downstream data size and alignment of distribution. We then propose a simple but effective method for fine-tuning that performs well under all setups.

A common concern among the reviewers was that fully finetuning the model is an oracle approach which we did not show in the paper. This is not true because the premise of CLIP-like models is their ultra-large-scale — over 150M params on over 400M examples — a scale which is difficult to match in downstream tasks, and hence, fine-tuning the full model is not just impractical but also too unstable. That being said, we’re pleased to report the additional experiments suggested by the reviewers including a fully finetuned baseline.

Below, we provide a summary of only the key questions and refer to individual replies for detailed rebuttal.

> [R2 # aQ8A, R3 # Vfao, R4 # Dgds] *"Full-model fine-tuning results"*
- We find that generally across both the low and high data settings, full-model finetuning is too unstable and catastrophically fails the majority of the time compared to the analyzed baselines. This is because CLIP has around 150M parameters and originally needed 400M examples during training. Compared to this, even our high-data downstream tasks have at most 100K examples.
- In the absolute best case, where we are able to fine-tune the full model, it results in performance slightly worse than zero-shot CLIP. We attribute this behavior to the difficulty of optimizing all of the parameters in CLIP and improving upon its initialization. Note that the parameter efficient methods we study don’t suffer from this problem.
- Results: https://drive.google.com/file/d/1evCSx7Aqg9S1QS3oMGuaIIX47n63hy_X/view?usp=sharing

> [R3 # Vfao] *Ablation of LayerNorm parameters*
- As requested by the reviewer, we provide an ablation of LayerNorm tuning where we consider only updating the LayerNorm parameters in the last K residual blocks of the vision and text transformers for K = [2, 4, 6, 8, 10, 12]. We consistently find that across all datasets and shots, increasing the number of residual blocks with tuned LayerNorm parameters results in improved performance up to K = 12. This indicates that all tuned LayerNorm parameters contribute to the final output representation — this is consistent with our proposed approach.
- Full results: https://drive.google.com/file/d/1YEZJruYsGH2DISIDCwhraArwl3EKCuwH/view?usp=sharing

> [R1 # GtTp, R2 # aQ8A, R3 # Vfao, R4 # Dgds] *Novelty and contribution concerns*

We address novelty and contribution concerns of several reviewers as follows:

- **Adaptation Methods (NLP vs Vision)**: The adaptation methods which we study have *previously been studied predominantly in NLP settings for language models* while we study them for classification tasks *in vision*. We believe this is an important contribution as large-scale pre-trained models like CLIP have become popular but are not easy to reproduce or retrain given their ultra large scale (150M params, 400M examples) for every downstream task. Furthermore, it’s unclear which fine-tuning methods to use in different situations.

- **CLIP as base model**: We choose CLIP for this analysis as it is one of a kind in scale and format among open-sourced models. We believe they are of interest to many researchers because of the enormous impact of CLIP. Since its release earlier this year, it has been used across robotics [CLIPort: Shridhar etal., CoRL 2021], videos [CLIP-It!: Narasimhan etal, 2021], image synthesis [StyleGAN-NADA: Gal etal 2021], and more.

- **Results and Conclusions**: First, in a vacuum, simple LayerNorm tuning is a strong yet simple baseline (and relatively model and task-agnostic) across a diverse set of downstream tasks. Second, combining it with other methods like linear probe or compactor networks can produce large gains over the baseline methods. We demonstrate this most clearly in the first two rows of Table 1 and our best performing results in Tables 3 and 4. Third, we note that the strong performance of linear probe with fine-tuned LN as initialization provides evidence towards the strength of LayerNorm tuning and the importance of joint vision-and-language training for strong visual representations.

>  [R3 # Vfao, R4 # Dgds] Title updated
- We have updated the title by taking a combination of changes proposed by reviewer # Vfao and # Dgds to narrow the scope of the paper as: "How to Adapt Your Large-Scale Pre-trained Model for Downstream Image Classification". For more details, see individual replies.

We have also updated the paper and supp.

---

### Decision · Program_Chairs · 2022-01-20

**Decision:**

Reject

**Comment:**

This paper investigates several fine-tuning methods for adapting the pretrained vison-language model CLIP to different downstream tasks.  They found that LayerNorm (Ba et al., 2016) is a rather effective and competitive approach and investigated different ways of combining LayerNorm-tuning with other adaptation methods. Reviewers generally agree that the technical novelty (combination of existing techniques) is limited and contribution is marginally significant. While some empirical results look interesting, the overall contribution is incremental. Overall, the paper has done a good job of thorough empirical evaluations, but the overall technical novelty and new empirical findings are not significant enough for publication at this conference.